# Exploring Balanced Feature Spaces for Representation Learning

**Bingyi Kang[1], Yu Li [2], Zehuan Yuan[3], Jiashi Feng[1]**
[1]National University of Singapore, [2]Institute of Computing Technology, CAS, [3]ByteDance AI Lab
kang@u.nus.edu,liyu@ict.ac.cn,yuanzehuan@bytedance.com,elefjia@nus.edu.sg

## Abstract

Existing self-supervised learning (SSL) methods are mostly applied for training representation models from artificially balanced datasets (*e.g.* ImageNet). It is unclear how well they will perform in the practical scenarios where datasets are often imbalanced w.r.t. the classes. Motivated by this question, we conduct a series of studies on the performance of self-supervised contrastive learning and supervised learning methods over multiple datasets where training instance distributions vary from a balanced one to a long-tailed one. Our findings are quite intriguing. Different from supervised methods with large performance drop, the self-supervised contrastive learning methods perform stably well even when the datasets are heavily imbalanced. This motivates us to explore the *balanced feature spaces* learned by contrastive learning, where the feature representations present similar linear separability w.r.t. all the classes. Our further experiments reveal that a representation model generating a balanced feature space can generalize better than that yielding an imbalanced one across multiple settings. Inspired by these insights, we develop a novel representation learning method, called $k$-positive contrastive learning. It effectively combines strengths of the supervised method and the contrastive learning method to learn representations that are both discriminative and balanced. Extensive experiments demonstrate its superiority on multiple recognition tasks, including both long-tailed ones and normal balanced ones. Code is available at https://github.com/bingykang/BalFeat.

## 1 Introduction

Self-supervised learning (SSL) has been popularly explored as it can learn data representations without requiring manual annotations and offer attractive potential of leveraging the vast amount of unlabeled data in the wild to obtain strong representation models (Gidaris et al., 2018; Noroozi & Favaro, 2016; He et al., 2020; Chen et al., 2020a; Wu et al., 2018). For instance, some recent SSL methods (Hénaff et al., 2019; Oord et al., 2018; Hjelm et al., 2018; He et al., 2020) use the unsupervised contrastive loss (Hadsell et al., 2006) to train the representation models by maximizing the instance discriminativeness, which are shown to generalize well across various downstream tasks, and even surpass the supervised learning counterparts in some cases (He et al., 2020; Chen et al., 2020a).

Despite the great success, existing SSL methods focus on learning data representations from the artificially balanced datasets (*e.g.* ImageNet (Deng et al., 2009)) where all the classes have similar numbers of training instances. However in reality, since the classes in natural images follow the Zipfian distribution, the datasets are usually imbalanced and show a long-tailed distribution (Zipf, 1999; Spain & Perona, 2007), *i.e.*, some classes involving significantly fewer training instances than others. Such imbalanced datasets are very challenging for supervised learning methods to model, leading to noticeable performance drop (Wang et al., 2017; Mahajan et al., 2018; Zhong et al., 2019). Thus several interesting questions arise: How well will SSL methods perform on imbalanced datasets? Will the quality of their learned representations deteriorate as the supervised learning methods? Or can they perform stably well? Answering these questions is important for understanding the behavior of SSL in practice. But these questions remain open as no research investigations have been conducted along this direction so far.

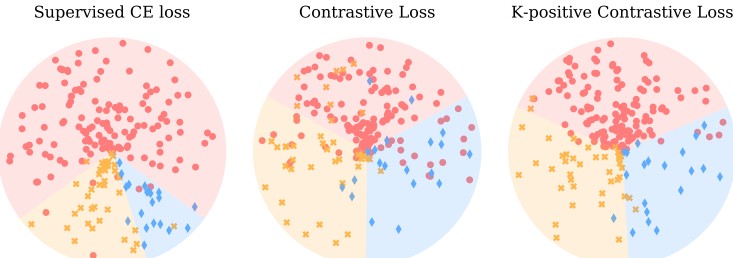

Figure 1: *Feature spaces learned with different losses given an imbalanced dataset.* The supervised cross-entropy (CE) learns a space biased to the dominant class. The space learned by unsupervised contrastive loss is balanced but less semantically discriminative. Our proposed $k$-positive contrastive loss learns a balanced and discriminative feature space. The shadow area ( ) indicates the decision boundary of each class.

Our work is motivated by the above questions to study the properties of data representations learned with supervised/self-supervised methods in a practical scenario. We start with two representative losses used by these methods, *i.e.*, the supervised cross-entropy and the unsupervised contrastive losses (Hadsell et al., 2006; Oord et al., 2018), and investigate the classification performance of their trained representation models from multiple training datasets where the instance distribution gradually varies from a balanced one to a long-tailed one. We surprisingly observe that, different from the ones learned from supervised cross-entropy loss where performance drops quickly, the representation models learned from the unsupervised contrastive loss perform stably well, no matter how much the training instance distribution is skewed to be imbalanced. Such a stark difference between the two representation learning methods drives us to explore why SSL performs so stably. We find that **using the contrastive loss can obtain representation models generating *a balanced feature space*** that has similar separability (and classification performance) for all the classes, as illustrated in Figure 1.

Such a balanced property of the feature spaces from SSL is intriguing and provides a new perspective to understand the behavior of SSL methods. We dig deeper into its benefits via a systematic study. In particular, since a pre-trained representation model is often used as initialization for downstream tasks (He et al., 2020; Newell & Deng, 2020; Hénaff et al., 2019), we evaluate and compare the generalization ability of the models that produce feature spaces of different balanced levels (or 'balancedness'). We find that **a more balanced model tends to generalize better across a variety of settings**, including the out-of-distribution recognition as well as the cross-domain and cross-task applications. These studies imply that feature space balancedness is an important but often neglected factor for learning high-quality representations.

Inspired by the above insights, we propose a new representation learning method, the $k$-positive constrastive learning, which inherits the strength of constrastive learning in learning balanced feature spaces and meanwhile improves the feature spaces' discriminative capability. Specifically, different from the contrastive learning methods lacking semantic discriminativeness, the proposed $k$-positive constrastive method leverages the available instance semantic labels by taking $k$ instances of the same label with the anchor instance to embed semantics into the contrastive loss. As such, it can learn representations with desirable balancedness and discriminativeness (Figure 1). Extensive experiments and analyses clearly demonstrate its superiority over the supervised learning and latest contrastive learning methods (He et al., 2020) for various recognition tasks, including visual recognition in both long-tailed setting (*e.g.*, ImageNet-LT, iNaturalist) and balanced setting.

This work makes the following important observations and contributions. (1) We present the first systematic studies on the performance of self-supervised contrastive learning on imbalanced datasets which are helpful to understanding the merits and limitations of SSL in practice. (2) Our studies reveal an intriguing property of the model trained by contrastive learning—the model can robustly learn balanced feature spaces—that has never been discussed before. (3) Our empirical analysis demonstrates that learning balanced feature spaces benefits the generalization of representation models and offer a new perspective for understanding deep model generalizability. (4) We develop a new method to explicitly pursue balanced feature spaces for representation learning and it outperforms the popular cross-entropy and contrastive losses based methods. We believe our findings and the novel $k$-positive contrastive method are inspiring for future research on representation learning.

## 2 RELATED WORKS

Self-supervised learning is a form of unsupervised learning. Recently there has been a surge of self-supervised data representation learning methods developed to alleviate the demand for manual annotations by mining free supervision information through specifically designed loss functions and pretext tasks. The contrastive loss measures the similarities of sample pairs in a feature space and is at the core of several recent SSL methods (Chen et al., 2020a;b; He et al., 2020; Chen et al., 2020c). Adversarial losses that measure the distribution difference are also exploited for self-supervised representation learning (Donahue et al., 2016; Doersch & Zisserman, 2017). A wide range of pretext tasks have been developed including image inpainting (Jenni & Favaro, 2018; Pathak et al., 2016), image colorization (Larsson et al., 2016; 2017), context prediction (Doersch et al., 2015), jigsaw puzzles (Carlucci et al., 2019; Noroozi & Favaro, 2016; Wei et al., 2019), rotation prediction (Gidaris et al., 2018). Though very successful, the behavior of SSL largely remains a mystery. Recently Wang & Isola (2020) analyze contrastive learning from the perspective of uniformity and alignment of learned representations. However investigations on the behavior of contrastive learning on imbalanced datasets are still absent. We present the first study on this problem and our investigation methodology is also applicable to other SSL methods.

In practice, the visual data usually follow a long-tailed distribution (Zipf, 1999; Spain & Perona, 2007), challenging supervised learning methods. Due to the imbalance in the number of training instances for different classes, conventional methods tend to perform much more poorly on instance-rare classes than on instance-rich ones. To alleviate this performance bias, existing approaches either re-balance the data distribution through sampling (Chawla et al., 2002; Han et al., 2005; Shen et al., 2016; Mahajan et al., 2018) or the loss for each class (Cui et al., 2019; Khan et al., 2017; Cao et al., 2019; Khan et al., 2019) by reweighting. Kang et al. (2020) first propose to decouple representation learning from classifier learning to boost performance, and demonstrate that learning good feature spaces is crucial for long-tailed recognition. Along this direction, SSP (Yang & Xu, 2020) is among the first methods that introduce SSL pretraining into learning the long-tailed recognition models. More specifically, instead of directly training a randomly initialized model from scratch as conventional supervised learning methods, SSP uses a model pretrained with SSL on the same dataset for initialization, which is observed to be able to to alleviate the label bias issue in imbalanced datasets and boost long-tailed recognition performance.

In contrast, we conduct a series of systematic studies to directly compare SSL with supervised learning on representation learning. We show that SSL can learn stably well feature spaces robust to the underlying distribution of a dataset. Moreover, inspired by our findings on the benefits of a balanced feature space for generalization, we introduce the $k$-positive contrastive learning method to explicitly pursue balancedness and discriminativeness for representation learning, which has been shown through experiments to benefit not only long-tailed recognition but also normal recognition tasks.

## 3 BALANCED FEATURE SPACES FROM CONTRASTIVE LEARNING

In this section, we systematically study the performance of representation models trained by SSL from a collection of training datasets with varying instance number distributions, in contrast with the models learned by supervised learning methods, to explore how SSL performs when the training datasets are not artificially balanced. Furthermore, we investigate the generalization performance of these learned representation models under multiple settings, in order to explore the relationship between the representation model's generalizability and the property of its learned feature space.

**Notations** We define the notations used in this paper. Representation learning aims to obtain a representation model $f_\theta$ that maps a sample $x_i$ into a feature space $V$ such that its corresponding representation $v_i \in V$ encapsulates desired features for target applications. Let $\mathcal{D}_{\text{rep-train}} = \{x_i, y_i\}$, $i = 1, \ldots, N$ be the dataset for training the representation model, where $y_i$ is the class label for sample $x_i$. Let $C$ denote the number of total classes and $n_j$ denote the number of instances within class $j$. We use $\{q_1, \ldots, q_C\}$ with $q_j = n_j/N$ to denote the discrete instance distribution over the $C$ classes. An *imbalanced* dataset has significant difference in the class instance numbers, *e.g.*, $q_1 \gg q_C$. We use a multi-layer convolutional neural network $f_\theta(\cdot) : x_i \mapsto v_i$ to implement

the representation model. The final classification prediction $\hat{y}$ is given by a linear classifier $\hat{y} = \arg\max[W^\top v + b]$, where $W$ denotes the classifier weight matrix and $b$ denotes the bias term.

## 3.1 METHODOLOGY OF OUR STUDY

**Representation learning methods**  Various loss functions have been developed for learning the representation model $f_\theta$ on the training dataset. Among them, the most popular one is the supervised cross-entropy (CE) loss:

$$\mathcal{L}_{\mathrm{CE}} = \frac{1}{N} \sum_{i=1}^{N} -\log p_{y_i}, \tag{1}$$

where $p_{y_i} = \mathrm{softmax}(W_{y_i}^\top v_i + b)$ is the normalized probability prediction of sample $i$ belonging to its ground truth class $y_i$. Using the semantic labels directly as supervision signal ($y_i$ in Equation 1), the representation model trained by the CE loss can have strong semantic discrimination ability but its generated feature space is easily biased by the imbalance of the training instance distribution—if some classes have significantly more training instances than the others, their data representations will occupy dominant portion of the feature space (Figure 1) and get higher classification accuracy than the instance-rare classes (Kang et al., 2020; Wang et al., 2017).

Different from the supervised learning ones, self-supervised learning methods adopt semantic-free loss functions to learn representations from unlabeled data (He et al., 2020; Gidaris et al., 2018). For example, the contrastive loss[1] (CL) (Oord et al., 2018) learns representations via maximizing the instance-wise discriminativeness:

$$\mathcal{L}_{\mathrm{CL}} = \frac{1}{N} \sum_{i=1}^{N} -\log \frac{\exp(v_i \cdot v_i^+/\tau)}{\exp(v_i \cdot v_i^+/\tau) + \sum_{v_i^- \in V^-} \exp(v_i \cdot v_i^-/\tau)}, \tag{2}$$

where $\tau$ is a temperature hyper-parameter, $v_i^+$ is a positive sample for the anchor instance $i$ (typically produced by data augmentation), $v_i^- \in V^-$ is the negative sample randomly drawn from the training samples excluding instance $i$. This contrastive loss encourages the feature representations from positive pairs to be similar, while pushing features from the sampled negative pairs apart.

We take the two loss functions ($\mathcal{L}_{\mathrm{CE}}$ and $\mathcal{L}_{\mathrm{CL}}$) as representatives to study how the representation models (and the corresponding feature spaces) trained with supervised/self-supervised methods are affected by the training instance distribution $\{q_1, \ldots, q_C\}$.

**Balancedness of feature spaces**  Since semantic labels are not involved in the contrastive loss (Equation 2), we hypothesize it may lead to representation models yielding feature spaces that are less biased by the imbalance of the training dataset, compared with the ones from the supervised loss (Equation 1). To verify this, we introduce a metric to characterize such an "unbiased" or "balanced" property of a feature space at first. A feature space $V$ is *balanced* if the representations $\{v_i\}$ from different classes within it have similar degrees of linear separability. As the linear separability degree of the representations is usually evaluated by the accuracy of a linear classifier over them (Vapnik, 2013), we follow this criterion to develop the balancedness metric. Specifically, let $a_1, \ldots, a_C$ denote the classification accuracy of a linear classifier $(W, b)$ over the representations $\{v_i\} \subset V$ from $C$ classes. We take the following uniformity of these accuracies as the *balancedness* of the feature space $V$:

$$\beta(V) \triangleq \frac{1}{C^2} \sum_{i,j}^{C} \exp\left(-\frac{|a_i - a_j|^2}{\sigma}\right), \quad \text{where } a_j = \frac{\#\{v_i | \hat{y}_i = j, y_i = j, v_i \in V\}}{\#\{v_i | y_i = j, v_i \in V\}}. \tag{3}$$

Here $\sigma$ is a fixed scaling parameter. This metric achieves its maximum when all the class-wise accuracies are equal, *i.e.*, there being no separability bias of the learned representations to any class. Note that this metric is developed to provide a quantitative measure of the balancedness of a feature space, but it has certain limitations such as it can be easily hacked. We leave developing more rigorous metric that can better characterize balanced feature spaces as future work.

---

[1]The term of *contrastive loss* has been used to refer to various loss functions over positive and negative samples. This work focuses on the specific form in Equation 2 that is widely used in modern SSL methods.

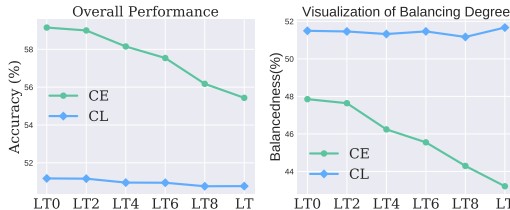
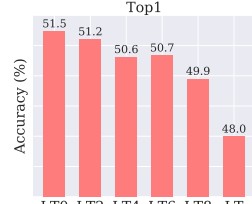
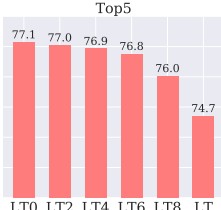

Figure 2: Classification accuracy (left) and balancedness (right) of the representations learned from cross-entropy (CE) loss and contrastive loss (CL) on datasets (LT0 to LT) with increasing imbalance.

Figure 3: Out-of-distribution generalization on ImageNet. Top 1 and Top 5 testing accuracy of the model are learned from datasets LT0 to LT that are increasingly more imbalanced.

**Experimental protocol**   We adopt a multi-stage protocol for learning and evaluating the feature spaces. (1) *Representation learning*: pre-train the representation model $f_\theta$ on the provided training set $\mathcal{D}_{\text{rep-train}}$ using the above training losses $\mathcal{L}_{\text{CE}}$ and $\mathcal{L}_{\text{CL}}$; (2) *Classifier learning*: train a linear classifier $(W, b)$ on top of $f_\theta$ with $\theta$ fixed using another training dataset $\mathcal{D}_{\text{train}}$[2] and the supervised CE loss; (3) *Representation evaluation:* evaluate the classification accuracy of the learned classifier on the test dataset $\mathcal{D}_{\text{test}}$ with the representations from $f_\theta$ and compute the above balancedness $\beta(V)$.

To thoroughly investigate sensitiveness of different representation learning methods to the imbalance level of training datasets, we construct six datasets from the long-tailed benchmark ImageNet-LT (Liu et al., 2019) ($\mathcal{D}_{\text{LT}}$) by varying its instance distribution $\{q_1, \ldots, q_C\}$ from a long-tailed one to a uniform one gradually, while keeping the total instance number similar. The generated datasets, denoted as $\mathcal{D}_{\text{LT0}}, \ldots, \mathcal{D}_{\text{LT8}}, \mathcal{D}_{\text{LT}}$ (which are increasingly more imbalanced), are used as $\mathcal{D}_{\text{rep-train}}$ for representation learning in the following experiments. See appendix for their details.

### 3.2   CONTRASTIVE LOSS HELPS LEARN BALANCED FEATURE SPACES

We first investigate classification performance of the representation models trained with the CE and CL losses on the above six datasets $\mathcal{D}_{\text{LT0}}, \ldots, \mathcal{D}_{\text{LT8}}, \mathcal{D}_{\text{LT}}$ that are increasingly more imbalanced. Since linear classifiers are easily biased by skewed training dataset distribution (Kang et al., 2020), it is necessary to eliminate the imblancedness of the evaluation datasets for reliable representation evaluation. Thus, we use the (balanced) training and test sets of ImageNet as $\mathcal{D}_{\text{train}}$ and $\mathcal{D}_{\text{test}}$ to learn classifiers and evaluate their classification accuracy, following the above protocol.

The results are summarized in Figure 2, from which we make an important observation: compared with the supervised cross-entropy loss, **the model trained with the unsupervised contrastive loss generates a more balanced feature space**, even in presence of highly imbalanced training instance distribution. As shown in Figure 2 (left), the classification accuracy of representation models learned with the CE loss drops quickly when the dataset becomes more imbalanced—the quality of representations from these models is very sensitive to the imbalance of training datasets. In contrast, the classification accuracy of CL-trained models remains stable even when the training dataset transits to a heavily long-tailed one. Such surprising performance robustness to imbalance of the training datasets implies that using contrastive learning can consistently learn balanced feature spaces. To see this, we also visualize the balancedness scores (Equation 3) of the learned feature spaces from CE and CL in Figure 2 (right). Even when the training set is heavily long-tailed, the feature spaces learned with CL loss are as highly balanced as the ones learned from a uniform training distribution, while the blancedness score of the feature spaces from CE loss is lower and drops quickly. Such a balanced feature space offered by CL loss is much desired for the representation learning in practice, where the training instance distribution is usually long-tailed. Certainly, using an unsupervised loss will sacrifice semantic discriminativeness of the representations, leading to the accuracy gap between the CE and CL-trained models.

### 3.3   MORE BALANCED REPRESENTATION MODELS GENERALIZE BETTER

The above studies reveal that the representation models trained with the contrastive loss can produce more balanced feature spaces. A natural question is what are the benefits from a balanced model for

---

[2]Note $\mathcal{D}_{\text{train}}$ and $\mathcal{D}_{\text{rep-train}}$ can be the same dataset as in recent SSL works (*e.g.*, He et al. (2020)).

Table 1: Results on Places365, VOC and COCO. $AP_{50}$ is the default metric for VOC, while $AP^{bb}$ and $AP^{mk}$ denote the bounding-box and mask AP for COCO respectively. Black / gray numbers correspond to results of the representation models trained on ImageNet-LT / ImageNet respectively. See appendix for complete results.

| | cross-domain | cross-task | | |
|---|---|---|---|---|
| | Places365 (Top1) | VOC ($AP_{50}$) | COCO ($AP^{bb}$) | COCO ($AP^{mk}$) |
| CE | 38.50 / 46.06 | 76.45 / 81.26 | 38.13 / 40.08 | 33.29 / 34.85 |
| CL | 41.24 / 46.16 | 78.19 / 82.28 | 39.67 / 40.41 | 34.73 / 35.14 |
| $\Delta_{CL,CE}$ | **+2.74** / +0.10 | **+1.64** / +0.02 | **+1.54** / +0.33 | **+1.44** / +0.29 |

recognition? Since a pre-trained representation model is often used to facilitate downstream tasks (He et al., 2020; Newell & Deng, 2020; Hénaff et al., 2019), we here conduct extensive experiments to study its potential benefits on model generalization performance under the following settings.

**Out-of-distribution generalization** We first study the relationship between the balancedness of representation models and their generalizability for recognizing new classes. To thoroughly evaluate performance of representation models with different balancedness, we evenly divide the 1,000 classes into two splits (500 vs. 500 classes) on ImageNet, referred to as the *source* and *target* split respectively. We use the subsets (corresponding to the source class split) of the above $\mathcal{D}_{LT0}, \ldots, \mathcal{D}_{LT8}, \mathcal{D}_{LT}$ datasets to construct six different $\mathcal{D}_{rep\text{-}train}$ for training the representation model $f_\theta$. To obtain models with different balancedness, we use the CE loss for training, since the above studies reveal using the CL loss will always produce models with similar balancedness (Figure 2). We use the subsets (corresponding to the target class split) of the training and test sets of ImageNet as $\mathcal{D}_{train}$ and $\mathcal{D}_{test}$ for classifier learning and evaluation, with the representation model fixed.

The testing performance of the models with different balancedness on the target classes is presented in Figure 3. It is observed that as the source dataset becomes increasingly more imbalanced (from $\mathcal{D}_{LT0}$ to $\mathcal{D}_{LT}$) and the corresponding models become more imbalanced, their generalization performance degrades correspondingly. Such a positive correlation between balancedness of the models and testing accuracy on the target classes clearly demonstrate that **more balanced representation models tend to generalize better** for recognizing unseen classes. More details and results about the out-of-distribution generalization studies are deferred to the appendix.

**Cross-domain and cross-task generalization** We then explore whether learning balanced representation models is able to benefit model's generalizability to new domains and tasks. We use the ImageNet-LT as $\mathcal{D}_{rep\text{-}train}$ to train the models with the CE and CL losses, obtaining imbalanced and balanced representation models respectively. For the cross-domain setting, we train a linear classifier on the Places365 dataset (Zhou et al., 2017) with the representation model fixed. From the results in Table 1, it can be clearly observed that the balanced representation model (from CL) surpasses the less balanced one (from CE) significantly, in terms of the top-1 accuracy (by 2.74%). For the cross-task setting, we take train/test splits of the PASCAL VOC (Everingham et al., 2010) and COCO (Lin et al., 2014) datasets as $\mathcal{D}_{train}/\mathcal{D}_{test}$ for evaluating detection performance. The results are given in Table 1. Again, the balanced model (from the CL loss) outperforms the less balanced one (from the CE loss) significantly (up to 1.64%). In comparison, the improvement from the CL-trained model over the CE-trained model is moderate (around 0.3%) when using the full ImageNet for training, as CE can learn a relatively balanced feature space from a balanced dataset. This clearly shows that the generalization performance boost for the cross-domain and cross-task settings brought from CL-trained models does not simply stem from using self-supervised pre-training, but indeed come from learning more balanced feature spaces.

## 4 LEARNING BALANCED FEATURE SPACES FOR RECOGNITION

The above studies demonstrate the representation models trained with the contrastive loss can generate balanced feature spaces showing strong generalizability. Here we explore how to effectively leverage these findings in practice. We introduce a new method that inherits the strength of the contrastive loss in learning balanced feature spaces and enhances the feature spaces' semantic discrimination capability simultaneously. We thoroughly study its superiority via two application cases, *i.e.*, the long-tailed recognition and pre-training representation models for downstream tasks.

### 4.1 K-POSITIVE CONSTRASTIVE LOSS

Though balanced, the feature spaces from contrastive learning have limited capability of semantic discrimination, as shown in Figure 2 (left). This is because the contrastive loss blindly encourages instance-level discrimiantiveness. Every two instances, even if they are from the same class, are forced to be apart from each other in the learned feature space. To embed semantic discriminativeness into the representations while maintaining desired balancedness, we develop a new method to leverage the provided semantic labels to adaptively compute the instance contrastive loss.

Concretely, given an anchor training instance $x_i$ with its semantic label $y_i$, our proposed method draws $k$ instances from the same class to form the positive sample set $V_{i,k}^+$, instead of only using its augmentation as in Equation 2. Thus, it gives a new loss called $k$-positive contrastive loss (KCL):

$$\mathcal{L}_{\text{KCL}} = \frac{1}{N(k+1)} \sum_{i=1}^{N} \sum_{v_j^+ \in \{\tilde{v}_i\} \cup V_{i,k}^+} - \log \frac{\exp(v_i \cdot v_j^+ / \tau)}{\exp(v_i \cdot \tilde{v}_i / \tau) + \sum_{v_j \in V_i} \exp(v_i \cdot v_j / \tau)}, \quad (4)$$

where $\tilde{v}_i$ is generated by augmenting $v_i$, $V_i$ is the current batch of examples excluding $v_i$, and $V_{i,k}^+ \subset V_i$ is a positive set containing $k$ instances randomly drawn from the same class as $v_i$. The proposed KCL loss purposely keeps the number of positive instances equal, which is crucial for balancing the learned feature spaces. It brings two benefits. First, it helps learn representations with stronger discriminative ability as it leverages the label information as supervised learning. Secondly, it uses the same number of instances (*i.e.*, $k$) for all the classes in positive pair construction which further balances the learned feature space. Note our proposed KCL is different from the supervised contrastive learning (Khosla et al., 2020) that leverages all the instances from the same class to construct the positive pairs, which cannot avoid the dominance of instance-rich classes in the representation learning. This is also evidenced by our following experiments on long-tailed recognition. In the following experiments, we choose $k = 6$ via validation and use ResNet50 as the backbone. Other hyper-parameter choices and implementation details are given in the appendix.

### 4.2 LONG-TAILED RECOGNITION

KCL provides feature spaces with desirable balancedness and semantic discriminativeness, which makes it naturally fit for addressing the challenges of long-tailed recognition, *i.e.*, severe performance bias to the instance-rich classes and poor generalization to the instance-rare classes (Mahajan et al., 2018; Zhong et al., 2019). Here we implement and evaluate KCL for long-tailed recognition, following the two-stage training strategy from Kang et al. (2020): 1) train the representation model with the KCL loss; 2) learn a linear classifier with cross-entropy loss and class-balanced sampling.

**Baselines** Besides well established state-of-the-arts, we consider following three kinds of baselines for justifying the advantages of KCL. (1) *Classifier balancing* methods, *i.e.*, $\tau$-norm and cRT (Kang et al., 2020), that re-train classifiers with class-balanced sampling as KCL but learn the representation models by supervised cross-entropy loss. Comparison with them helps understand the effectiveness of learning balanced features in long-tailed recognition. (2) Methods that train the representation model and classifier jointly with cross-entropy loss (SL) and various data re-sampling strategies, including instance-balanced (SL-i), class-balanced (SL-c), progressively-balanced (SL-p) and square-root re-sampling (SL-s) (Kang et al., 2020). Comparison with them will show advantages of KCL over these data-enriching strategies in feature space balancing. (3) *A full-positive variant of KCL*, named full-positive contrastive learning (FCL), that uses all the available same-class samples in the current batch to construct positive pairs for computing the contrastive loss, which is similar to the supervised contrastive learning (Khosla et al., 2020). Comparing KCL with FCL will show the benefits of keeping the number of positive samples equal for all the anchor instances in KCL.

**Results** We evaluate KCL and compare it with the above strong baselines on two large-scale benchmark datasets, ImageNet-LT (Liu et al., 2019) and iNaturalist 2018 (iNatrualist, 2018). For comprehensive evaluation, following (Liu et al., 2019), we split the classes of ImageNet-LT into many-shot ($>$100 images), medium-shot (20$\sim$100 images) and few-shot ($<$20 images) groups. The results are summarized in Tables 2 and 3 respectively, along with the balancedness of these methods on ImageNet-LT in Figure 4. We make the following observations.

*More balanced feature spaces give better performance.* We first compare KCL with cRT and $\tau$-norm, the latest state-of-the-arts with feature spaces learned by supervised cross-entropy loss and

Table 2: ImageNet-LT results

| Method | Many | Medium | Few | All |
|---|---|---|---|---|
| OLTR (Liu et al., 2019)[a] | 35.8 | 32.3 | 21.5 | 32.2 |
| Joint (SL-i) (Kang et al., 2020) | 64.9 | 35.2 | 6.8 | 42.5 |
| $\tau$-norm (Kang et al., 2020) | 56.6 | 44.2 | 27.4 | 46.7 |
| cRT (Kang et al., 2020) | 58.8 | 44.0 | 26.1 | 47.3 |
| FCL | 61.4 | 47.0 | 28.2 | 49.8 |
| KCL | 61.8 | 49.4 | 30.9 | **51.5** |

[a]Reproduced by re-running their code with ResNet50.

Table 3: iNaturalist 2018 results

| Method | Top1 |
|---|---|
| CB-Focal (Cui et al., 2019) | 61.1 |
| LDAM (Cao et al., 2019) | 64.6 |
| LDAM+DRW (Cao et al., 2019) | 68.0 |
| cRT (Kang et al., 2020) | 65.2 |
| $\tau$-norm (Kang et al., 2020) | 65.6 |
| BBN (Zhou et al., 2020) | 66.3 |
| FCL | 66.4 |
| KCL | **68.6** |

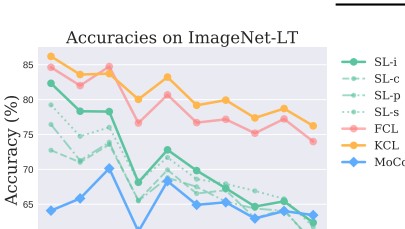

Figure 4: Comparison of different methods on their feature space balancedness (left) and class-wise accuracy (right). Here the linear classifier is fine-tuned on the full ImageNet for representation evaluation (see Sec. 3.2).

thus less balanced as demonstrated in Sec. 3. Compared with them, KCL improves the overall accuracy by a large margin (4.2% on ImageNet-LT and 3% on iNaturalist), demonstrating the importance of learning more balanced feature spaces for long-tailed recognition.

*KCL is more effective at learning balanced and discriminative feature spaces.* Data re-sampling is widely used as a straightforward approach to alleviate performance bias for long-tailed recognition (Kang et al., 2020). We compare the feature space balancedness of KCL and SL methods with different data re-sampling strategies on ImageNet-LT in Figure 4. Clearly, data re-sampling cannot effectively improve balancedness of the feature space as KCL. Besides data re-sampling, Figure 4 also shows the balancedness of the feature spaces learned by the latest contrastive learning method MoCo (He et al., 2020) on ImageNet-LT. MoCo can balance the feature space but has lower accuracy, due to the lack of semantic discriminativeness in the learned feature space. KCL performs the best, demonstrating its effectiveness at learning both balanced and discriminative feature spaces.

*Equalizing the number of positive instances in KCL is important.* To further justify the design of KCL loss in keeping the number of positive instances to be equal, we compare it with its variant FCL. From Tables 2, 3 and Figure 4, though FCL outperforms other baselines, its performance is inferior to KCL, in terms of both the overall accuracy and the balancedness of the learned feature spaces. Equalizing the number of positive instances as KCL is crucial for learning balanced feature spaces and improving recognition performance.

### 4.3 PRE-TRAINING REPRESENTATION MODELS FOR DOWNSTREAM TASKS

The effectiveness of KCL is not limited to the cases where training datasets are imbalanced. In this section, we study KCL as a general representation learning method, *i.e.*, we apply KCL for pre-training a representation model on balanced datasets which is later fine-tuned for downstream tasks, including the out-of-distribution (OOD) recognition and detection.

**Out-of-distribution Generalization** Similar to Sec.3.3, we evenly divide the 1000 classes in ImageNet into two splits, use one split to learning representation backbone and the thoer one to learn a linear classifier with the backbone fixed. We adopt two different splitting strategies. *Split-overlap* (split with semantic overlap) allows the classes within the two splits to share the same super class (*e.g.*, *dog* and *wolf* from *canidae* are put into different splits) in the ImageNet ontology. As such, though the target classes are all novel to the representation model, some of their attributes have been seen by the model before from the source classes. In contrast, *Split-independent* (split without semantic overlap) strictly avoids classes from the same super classes to be distributed into different splits. *Split-independent* presents a more challenging case for model's generalization ability as all the target classes (and attributes) to recognize are novel.

Table 4: *OOD generalization results (top-1 accuracy) on balanced datasets.* We use the *source* classes of origninal ImageNet to learn representation network (ResNet50), and use the *target* classes and *all* classes respectively to learn linear classifiers for evaluation.

|  | Split-overlap | | | Split-independent | | |
|---|---|---|---|---|---|---|
|  | source | target | all | source | target | all |
| CE | 81.2 | 70.7 | 67.2 | 82.8 | 50.3 | 62.4 |
| CL | 67.0 | 60.1 | 58.3 | 68.2 | 54.8 | 58.2 |
| KCL | 81.4 | 74.8 (+4.1) | 70.8 (+3.6) | 83.2 | 58.1 (+3.3) | 67.2 (+4.8) |

The generalization performance comparsion of the representation models with different methods is given in Table 4. When the source classes and target classes share similar semantics (on *split-overlap*), the CE-trained model surpasses the CL-trained model on both the source and target classes. But when looking into the generalization gap (*i.e.*, the difference between the source and target accuracy), the CL-trained model suffers larger generalization gap than the CL-trained model (10.5 vs. 1.8). When there is not semantic overlap between source and target (on *split-independent*), the CL-model outperforms CE-model on the target classes by 4.3% with much smaller generalization gap (13.4 v.s. 32.5). By comparing the "full" performance from *split-overlap* to *split-independent*, one can observe that CL loss performs consistently well (58.3 and 58.2), but CE drops as large as 5%. This implies that CL is robust to imbalanced training distribution used for representation learning, while CE is extremely sensitive to it. These results clearly demonstrate *the consistent superiority of balanced representation learning in terms of generalization* for various training dataset distribution. Notably, our proposed KCL loss surpasses both CE and CL loss on all the four different settings by a large margin (more than 3 points). These results clearly demonstrate that KCL is able to learn a balanced and discriminative feature space, and balancedness is a general property that benefits both balanced and imbalanced datasets.

Table 5: Comparison of different representation learning methods for the downstream tasks.

|  | repr | VOC ($AP_{50}$) | COCO ($AP^{bb}$) | COCO ($AP^{mk}$) |
|---|---|---|---|---|
| SL | 76.6 | 81.26 | 40.08 | 34.85 |
| MoCo | 60.6 | 81.28 | 40.41 | 35.15 |
| KCL | 76.8 | **82.32** | **40.79** | **35.45** |

**Cross-domain and cross-task generalization** In this part, we first pretrain a model on ImageNet then further finetune it for downstream object detaction tasks (including PASCAL VOC and COCO). Note we aim to study the generalizability of KCL as a representation learning method, rather than aiming at state-of-the-art performance. Hence we compare it with the vanilla supervised cross-entropy loss (SL) and MoCo (which the KCL is built on) (He et al., 2020). The results are summarized in Table 5. We also evaluate the discrminativeness of the learned representations (the "repr" in the table) from their classification accuracy by learning a linear classifier on the pretraining datasets. Clearly, KCL outperforms SL and MoCo for both downstream tasks. This is because KCL learns more balanced feature spaces than SL with similar discriminativeness, and learns more discriminative features than MoCo. For more results, Please refer to the appendix .

## 5 CONCLUSIONS

This work piloted studies on performance of the self-supervised learning methods for imbalanced datasets, and made several intriguing findings. At the heart of these findings is the balanced feature space, which is identified to be an inherent property of the representations learned by the contrastive learning and bring stronger generalizability. It provides a new perspective for understanding the behavior of the contrastive learning. This work further developed a new representation learning method to leverage the benefits of balanced feature spaces. We believe the findings and method developed here are inspiring for the future research on representation learning. However, theoretical understandings on balanced feature spaces are not mature yet and worthy of future exploration.

## ACKNOWLEDGEMENT

We would like to express our deepest gratitude to Saining Xie for his comments and suggestions throughout this project and the writing of the paper, to Yu Sun for his insightful discussion at the begining of this project.

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

## A   IMPLEMENTATION DETAILS

**Representation Learning**   For supervised cross-entropy (CE) loss, we adopt the standard PyTorch distributed training implementation[3]. For unsupervised contrastive loss (CL), we use the official implementation[4] of MoCo (He et al., 2020) with default hyper-parameters. Our $k$-positive contrastive loss is implemented based on MoCo by randomly selecting $k$ positive examples from the key memory for each of the query example. When KCL is applied for a long-tailed dataset, there might be less than $k$ positive examples in the memory for some of the classes. In such cases, we use all the positive examples. Besides, we keep all the hyper-parameters (*e.g.*, data augmentation, learning rate and batch size) the same as supervised learning. We only carefully tune the number of training epochs for KCL to make it achieve similar performance as supervised learning on a balanced dataset (full Imagenet) for fair comparison (see Table 6). As a result, throughout the paper our KCL is trained for 200 epochs while its supervised counterpart is trained for 90 epochs. This is reasonable as contrastive learning usually takes much longer to converge (He et al., 2020). We are using $k = 6$ for KCL throughout the paper, which is carefully tuned on the validation set of ImageNet-LT, as shown in Fig. 5. The detailed hyper-parameters of different loss functions are given in Table 7.

Table 6: Results on full ImageNet.

|  | Top1 | Top5 |
|---|---|---|
| CE (90 epochs) | 76.616 | 93.090 |
| SCL (200 epochs) | 76.976 | 92.972 |
| KCL (200 epochs) | 76.814 | 92.936 |

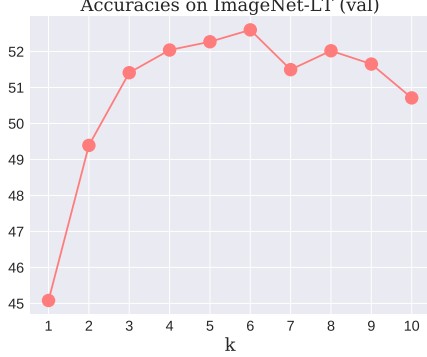

Figure 5: Validation accuracies of KCL on ImageNet-LT as the value of $k$ varies.

| $hps$ | CE | CL | KCL |
|---|---|---|---|
| epochs | 90 | 200 | 200 |
| batch size | 256 | 256 | 256 |
| learning rate | 0.1 | 0.03 | 0.1 |
| learning rate schedule | cosine | step | cosine |
| data augmentation | default | moco v1 | default |
| memory size | - | 65536 | 65536 |
| encoder momentum | - | 0.999 | 0.999 |
| feature dimension | - | 128 | 128 |
| softmax temperature | - | 0.07 | 0.07 |
| $k$ | - | - | 6 |

Table 7: Hyper-parameters used by different loss functions. "default" means the standard data augmentation strategies used by supervised learning.

**Classifier Learning**   We need to train linear classifiers in two cases. 1) *representation evalauation* (Sec. 3 and Figure 4). For CE-learned representations, we train a linear classifier using the same parameters as representation learning in Table 7 with a smaller number of epochs (10). For CL and KCL, we adopt the classifier training protocol introduced by MoCo (He et al., 2020) with default hyper-parameters (*i.e.*, learning rate 30 and weight decay 0). 2) *Long-tailed recognition* (Sec. 4.2). We obtain a re-balanced classifier for CE representations following Kang et al. (2020), and adopt the MoCo classifier learning strategy with class-balanced sampling by setting the learning rate to 10 on ImageNet-LT and 30 on iNaturalist 2018.

**Detection model training**   We use exactly the same setting and evaluation metrics as He et al. (2020). R50-C4 backbone is used with BN tuned. The image is rescaled to [640, 800] during training and 800 at inference. All layers are fine-tuned end-to-end with batch size = 16. For Pascal VOC, we train Faster R-CNN (Ren et al., 2015) on `trainval07+12` set with 24k schedule and evaluate on `test07` set. For COCO, we train Mask R-CNN (He et al., 2017) on `train2017` set with ×2 schedule and evaluate on `val2017` set.

---

[3]https://github.com/pytorch/examples/tree/master/imagenet
[4]https://github.com/facebookresearch/moco

## B  DATASET CONSTRUCTION

**Datasets for studying balancedness of feature spaces**  We here explain the details on the construction of the series of datasets used in our study in Sec. 3.

In particular, we take the standard long-tailed training set from ImageNet-LT (Liu et al., 2019) whose instances follow the Pareto distribution as the base dataset, denoted as $\mathcal{D}_{\text{LT}}$. We vary its training instance distribution $\{q_1, \ldots, q_C\}$ gradually to obtain different datasets as follows,

$$n_j = \left\lfloor N_{\mathcal{D}} \times \frac{q_j^\alpha}{\sum_k q_k^\alpha} + \frac{1}{2} \right\rfloor, \tag{5}$$

where $N_{\mathcal{D}}$ is the total number of training instances in $\mathcal{D}_{\text{LT}}$, $\alpha \in [0, 1]$ controls the dataset balancedness. When $\alpha = 0$, it corresponds to a fully balanced dataset; when $\alpha = 1$, it becomes a heavy long-tailed ones. In total, we generated 6 datasets with $\alpha \in \{0, 0.2, 0.4, 0.6, 0.8, 1.0\}$, denoted as $\mathcal{D}_{\text{LT0}}, \ldots, \mathcal{D}_{\text{LT8}}, \mathcal{D}_{\text{LT}}$ respectively, as different examples of $\mathcal{D}_{\text{rep-train}}$ for representation learning. The detailed statistics and visualization of the datasets $\mathcal{D}_{\text{LT0}}, \ldots, \mathcal{D}_{\text{LT8}}, \mathcal{D}_{\text{LT}}$ are summarized in Table 8 and Fig. 6.

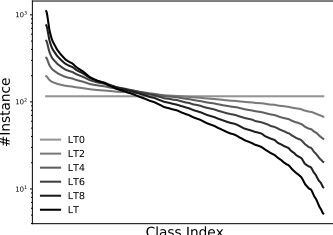

Figure 6: Training instance number distributions of the datasets we use in our empirical studies.

| Dataset | Max | Min | Total |
|---------|-----|-----|-------|
| $\mathcal{D}_{\text{LT0}}$ | 115 | 115 | 115,000 |
| $\mathcal{D}_{\text{LT2}}$ | 204 | 67 | 115,885 |
| $\mathcal{D}_{\text{LT4}}$ | 343 | 37 | 115,801 |
| $\mathcal{D}_{\text{LT6}}$ | 553 | 20 | 115,836 |
| $\mathcal{D}_{\text{LT8}}$ | 857 | 10 | 115,852 |
| $\mathcal{D}_{\text{LT}}$ | 1,280 | 5 | 115,846 |

Table 8: Dataset statistics on training instance numbers, including the maximal and minimal instance number per class and the total number.

**Datasets for generalizability studies**  We carefully choose the proper datasets to construct the $\mathcal{D}_{\text{train}}$ and $\mathcal{D}_{\text{test}}$ for evaluating generalizability of the representation models under multiple settings. The choices are summarized in Table 9.

Table 9: Summary on used datasets for representation model pre-training and evaluation in our studies.

| study | $\mathcal{D}_{\text{rep-train}}$ | $\mathcal{D}_{\text{train}}$ | $\mathcal{D}_{\text{test}}$ |
|-------|------------------|-------------|------------|
| Balancedness (Sec. 3.2) | $\mathcal{D}_{\text{LT0}}, \ldots, \mathcal{D}_{\text{LT8}}, \mathcal{D}_{\text{LT}}$ | ImageNet (train) | ImageNet (val) |
| Out-of-distribution (Sec. 3.3) | source split of above | target split of above | |
| Cross-domain (Sec. 3.3) | ImageNet-LT | Places 365 (train) | Places 365 (test) |
| Cross-task (Sec. 3.3) | ImageNet-LT | VOC/MSCOCO (train) | VOC/MSCOCO (test) |

## C  ADDITIONAL RESULTS ON MODEL GENERALIZATION PERFORMANCE

**Cross-domain and Cross-task Generalization**  We evaluate the generalization ability of the representation models trained on the balanced full ImageNet datasets, for cross-domain and cross-task applications. The results are given in Table 10 (cross-domain) and Tables 11 and 12 (for detection) respectively. From Table 10, when the training datasets are balanced, the models trained with CL and CE achieve comparable performance. While when the training datasets are not balanced, the CL model significantly outperforms the CE model (Table 1). This demonstrates that the CL loss can consistently produce balanced representation models and the model generalization performance can indeed benefit from being balanced.

Similar conclusion can be drawn for the cross-task generalization. From Tables 11 and 12, when training the model on the full ImageNet dataset, using self-supervised CL loss can produce the model performing slightly better than using the supervised CE loss. On PASCAL VOC, the performance

advantage is as marginal as 0.02% in $AP_{50}$. In contrast, when training the model on the ImageNet-LT dataset, using CL loss can boost the model performance over using the CE loss much more significantly. The improvement is as large as 1.64% in $AP_{50}$. Thus the performance benefit on the generalization to detection brought by CL does not simple stem from using self-supervised pre-training, but indeed come from learning more balanced feature spaces.

Similar to OOD generalization, the model learned with KCL gives better downstream performance on both VOC and COCO, which mean enforcing feature space balancedness with KCL is indeed able to help learning better representation.

Table 10: Results on Places 365. The encoder is trained on ImageNet.

|    | Top1 | Top5 |
|----|------|------|
| CE | 46.06 | 77.11 |
| CL | 46.16 (+0.1) | 76.27 (-0.84) |

Table 11: Object detection Results on PASCAL VOC. The representation model is trained on ImageNet and ImageNet-LT. We report results in $AP_{50}$: VOC metric; AP: COCO-style metric.

|     | ImageNet | | | ImageNet-LT | | |
|-----|----------|------|------|-------------|------|------|
|     | $AP_{50}$ | AP | $AP_{75}$ | $AP_{50}$ | AP | $AP_{75}$ |
| CE  | 81.26 | 53.66 | 59.19 | 76.45 | 48.53 | 51.01 |
| CL  | 81.28 (+0.02) | 56.10 (+2.44) | 62.71 (+3.52) | 78.19 (**+1.64**) | 51.52 (**+2.99**) | 56.48 (**+5.47**) |
| KCL | 82.32 (**+1.06**) | 55.51 (**+1.85**) | 62.05 (**+2.86**) | 79.70 (**+3.25**) | 52.63 (**+4.10**) | 57.89 (**+6.88**) |

Table 12: Object detection Results on COCO. The representation model is trained on ImageNet and ImageNet-LT. We report results in bounding-box AP ($AP^{bb}$) and mask AP ($AP^{mk}$).

|            |     | ImageNet | | | ImageNet-LT | | |
|------------|-----|----------|------|------|-------------|------|------|
|            |     | AP | $AP_{50}$ | $AP_{75}$ | AP | $AP_{50}$ | $AP_{75}$ |
| $AP^{bb}$  | CE  | 40.08 | 59.76 | 43.29 | 38.13 | 57.38 | 41.15 |
|            | CL  | 40.41 (+0.33) | 60.05 (+0.29) | 44.09 (+0.80) | 39.67 (**+1.54**) | 59.40 (**+2.02**) | 42.73 (**+1.58**) |
|            | KCL | 40.79 (**+0.78**) | 60.63 (**+0.87**) | 43.99 (**+0.70**) | 39.43 (**+1.30**) | 59.08 (**+1.70**) | 42.56 (**+1.41**) |
| $AP^{mk}$  | CE  | 34.85 | 56.60 | 37.02 | 33.29 | 54.24 | 35.38 |
|            | CL  | 35.14 (+0.29) | 56.88 (+0.28) | 37.56 (+0.54) | 34.73 (**+1.44**) | 56.07 (**+1.83**) | 37.13 (**+1.75**) |
|            | KCL | 35.45 (**+0.60**) | 57.40 (**+0.80**) | 37.80 (**+0.78**) | 34.38 (**+1.09**) | 55.81 (**+1.57**) | 36.39 (**+1.01**) |

