# OpenReview forum: "Exploring Balanced Feature Spaces for Representation Learning"
_ICLR.cc/2021/Conference — ICLR 2021 Poster_

### Official Review · AnonReviewer4 · 2020-10-23
**nice paper but the key discovery was published somewhere else**

**Rating:** 5
**Confidence:** 5

**Review:**

Summary:

- This paper made a key observation that the self-supervised contrastive learning methods perform stably well even when the datasets are heavily imbalanced. As for the method this paper proposes a classed-balanced version of supervised contrastive loss (Khosla et al., 2020). Extensive experiments demonstrate its superiority on multiple recognition tasks.

Pros:
- The paper is well written with nice figures.
- The paper conducted experiments on extensions other than image classification, which provides some useful insights.

Cons:
- The advantage of using self-supervised learning to learn a representation to combat label bias issue in imbalanced problems has been discovered and validated by a previous work [1]. I understand that [1] was accepted just a few days before the DDL of ICLR. Unfortunately I still have to devalue the contribution of this paper as it seems to me that the major contribution of this paper relies on the finding that SSL can help to alleviate the issue of label bias. In terms of large-scale experiments on ImageNet-LT and iNatualist, it seems that this paper has no advantage over [1].
- It seems the runtime of the proposed method is much worse than supervised based methods.

Additional Questions and Concerns:
- Why does the author use cosine lr schedular?
- How to interpret accuracys on imagenet in fig. 4? Why is there only 10 classes?

[1] Yang, Yuzhe, and Zhi Xu. "Rethinking the Value of Labels for Improving Class-Imbalanced Learning." NeurIPS 2020

----
post-rebuttal update

I appreciate the discussions between the authors. I plan to keep my original score, for the reason that, at least in my point of view, the difference of the two methods is subtle and it is not clear whether the subtle difference results in drastic improvement.

---

> ### Author Response · Authors · 2020-11-12
> **Response to AnonReviewer4: Our  studies and findings are novel and different from[3]**
>
> **A1.** *[The main discovery has been discovered and validated by a previous work.]*
>
> - First, our findings are different from the ones in [3]. The discovery made in [3] is “the advantage of using self-supervised learning to learn a representation to combat label bias issue in imbalanced problems.” However, our finding is: **unsupervised contrastive loss is able to learn balanced feature spaces, which is not only a property that benefits long-tailed recognition but also gives better generalization ability for normal representation learning models.** As evidenced in Sec. 4.3, when testing on fully balanced datasets (e.g. ImageNet), a more balanced feature produced by our proposed KCL method beats supervised learning on downstream tasks.  None of these have been discussed in [3].
>
> - Second, we would like to provide a more detailed discussion of [3].  [3] draws the conclusion that SSL pretraining is able to overcome label bias issues mainly by the following contrast experiments. They pretrain a model using SSL then further train it using supervised learning, resulting in a final model (refered as ssl model in the following). Then the ssl model is directly compared to a baseline model, which is directly trained with supervised learning.
>
>     The ssl model is trained with longer time (200 epochs with ssl loss, and 90 epochs with CE loss), than the baseline model (90 epochs with CE). Therefore, **the performance improvement observed by [3] may come from longer training, since training longer is shown to be very effective for long-tailed recognition in [4].** To exclude such a factor and make a fair comparison, we conduct experiments with two additional baselines: the “ssl model” is firstly trained with supervised learning for 90 epochs, then further trained with supervised learning for another 90 epochs.; another baseline model is simply trained for a longer time (200 epochs). The results are given as follows,
>
>     |Method  |                                        Many    |  Medium    |   Few   |    All  |
> | ----------  | ---------------------------------------- | --------------  | -------- | -------|
> |Baseline model (90 epochs)  |   59.735 |  45.113 | 27.082 | 48.110 |
> |Baseline model (200 epochs)  | 61.382	| 48.486 | 32.890 | 51.174  |
> | ssl model (200+90 epochs)   |   60.670 | 48.392 | 31.493 | 50.652  |
> |sl model (90+90 epochs)        |   60.078  | 47.684 | 31.082 | 50.032  |
>
>     Note that all results are given using tau-normalization proposed in [4]. **The results clearly show that ssl pretraining gives no additional benefits compared to supervised learning.** Moreover, we also tried to train a model by optimizing supervised loss and SSL loss together. The results show that the learned model is negatively affected as explained in answer 4 to reviewer #1. **All these results together prove that our proposed KCL loss is the first successful practice to combine the benefits of supervised learning and self-supervised learning.**
>
> - Third, besides the above discovery, we also make the following contributions which should not be neglected: (1) We are the first to show the performance of SSL is robust to dataset imbalance by studying SSL on various imbalanced datasets. (2) We propose a new concept of “balancedness” for feature space and demonstrate such property is important for both classification performance and generalization to different distributions and tasks. (3) The proposed method, KCL, is novel. It learns balanced and discriminative features, and outperforms CL and SL on both long-tailed datasets (Sec. 4.2) and balanced datasets (Sec. 4.3). We sincerely hope our contributions can be fairly treated.
>
> [3] Yang, Yuzhe, and Zhi Xu. "Rethinking the Value of Labels for Improving Class-Imbalanced Learning." NeurIPS 2020
>
> [4] Kang, Bingyi, et al. "Decoupling representation and classifier for long-tailed recognition." ICLR 2020.
>
> **A2.** *[The runtime of the proposed method is much worse than supervised based methods. ]*
>
> The proposed method needs longer training epochs because the contrastive loss takes much longer to converge, as explained in Appendix A. The same criticism can be directed to all the SSL methods in the literature, including [3]. However, our method enjoys the advantage of learning balanced and discriminative features,  which is demonstrated to be crucial for imbalanced classification problems.
>
>
> **A3.** *[Why cosine lr schedule?]*
>
> For MoCo, we use the original step decay strategy. For our proposed KCL method, we adopt the same lr scheduler used in supervised learning for a fair comparison, as explained in Appendix A.
>
> **A4.** *[How to interpret accuracies on imagenet in fig. 4? Why are there only 10 classes?]*
>
> We will add y-axis in the next version for Fig.4.  Actually, there are 1000 classes. For clearer visualization, we divide them into 10 bins according to their training instance numbers. Each dot in the figure represents the average accuracy of the corresponding bin.

---

> > ### Public Comment · ~Yuzhe_Yang1 · 2020-11-16
> > **Disagree with your claims on our paper**
> >
> > Hi, I’m the author of [3], and I happened to come across this paper. Since the discussions are related to our work, we read the reviews and would like to provide our perspectives on the author responses. First, I would like to thank Reviewer4 for pointing out our work, and indeed, we believe this submission is very much relevant to our paper. While the paper itself may have some merits, I however, found that the responses by the authors on [3] are problematic, and hence strongly disagree with the claims on our paper.
> >
> > To begin with, while our paper was just accepted to NeurIPS, it was made available ~4 months before ICLR deadline. Given it is highly related, we believe that it is necessary for the authors to discuss our paper in a later version to give the audience a complete picture about the progress of this field.
> >
> > Most importantly, the claims made by the authors on our paper are not true.
> > > the performance improvement observed by [3] may come from longer training... To exclude such a factor and make a fair comparison...
> >
> > First of all, our proposed self-supervised pre-training (SSP) aims to learn a better initialization. Then, in its second training stage (i.e., with supervised learning), it performs exactly what the baseline does. Therefore, a fair comparison should be as follows: in order to show SSP is useful, one should fix all the training parameters in the second (normal) training stage, and compare the performance with/without SSP. To be more precise, here the control variable is whether you use (self-supervised) pre-trained weights or not, i.e., the only difference is starting from a pre-trained network rather than random. Indeed, your experiments just directly demonstrate the effectiveness of SSP --- if you compare 1st row with 3rd/4th rows, there are clear improvements, given the same training epochs in the normal training stage (i.e., 90). To fairly compare with the 2nd row (baseline with 200 epochs), one should simply train also for 200 epochs as the baseline in the supervised training stage. To validate this argument, we quickly ran a simple experiment following your setting, but just with longer training time in supervised training stage (i.e., 200+200). The result is as follow:
> >
> > |Method|	Many|	Medium|	Few|	All|
> > | --------- |-----------|-------------|-------------|-------------|
> > |SSP (200+200)	|62.77|	48.79|	33.19|	51.90|
> >
> > To summarize, all these results together verify that, given the same training setup for baseline models, adding SSP consistently improves the result. This demonstrates that your claim
> > > The results clearly show that ssl pretraining gives no additional benefits compared to supervised learning
> >
> > is simply incorrect and misleading. Furthermore, we note that thorough experiments are needed before drawing a conclusion. Indeed, in our paper, we have performed an in-depth exploration of self-supervision on imbalanced learning over several dimensions: (1) different baseline methods, (2) different datasets, and (3) different self-supervised methods. We have shown consistent improvements regardless of what baseline methods one uses.
> >
> > Correspondingly, the claim
> > > All these results together prove that our proposed KCL loss is the first successful practice to combine the benefits of supervised learning and self-supervised learning
> >
> > is also not true. In fact, the claim itself can be problematic. First, the technique in this paper is essentially a variant of _supervised_ contrastive loss, which also requires label information in the first training stage. Therefore, this is essentially not an exact self-supervised learning technique. Second, to emphasize again, our paper has extensively tested the benefits of self-supervision over (1) different base methods, (2) different datasets, and (3) different self-supervised methods. The consistent improvements already confirm the successful practice of combining supervised learning and SSL. Thus, the argument here is not correct.
> > Consequently, the claim
> > > We are the first to show the performance of SSL is robust to dataset imbalance by studying SSL on various imbalanced datasets
> >
> > is not true and highly misleading.
> >
> > Finally, as Reviewer4 also noted, there is no performance advantage over our SSP method. To sum up, the claims that the current paper is the first to utilize SSL or have substantial improvements over SOTA, are not valid, and should not be considered as the main contributions of this paper.
> >
> > While I acknowledge the authors' efforts on the interesting findings, we would like to emphasize that the contributions of past works should also be fairly treated and acknowledged, and hence, we strongly disagree with the incorrect and misleading claims about our paper in the author responses.
> >
> > ------------
> > For a minor note, the table presented in the author responses is a bit strange --- it seems to be inconsistent with what was reported in the paper (e.g., $\tau$-norm 46.7 in Table 2, while 48.1 here), which may indicate potential errors.

---

> > > ### Author Response · Authors · 2020-11-18
> > > **Your clarification further confirms our claims: Part 1**
> > >
> > > We are happy to see the authors of [3] attend the discussion.
> > >
> > > We did not notice [3] until AR4 pointed us to it. Though per ICLR policy, it is not required to include unpublished papers or the papers published near to submission deadline,  we are definitely happy to add discussions on [3] in our future revision.
> > >
> > > First of all, we would like to emphasize again: **though our submission and [3] both involve self-supervised learning (SSL) for long-tailed recognition, our work significantly differs from [3] in motivation, methodology, and experiments.**
> > >
> > > - ***Motivation.***  We aim to study whether the good performance (observed on ImageNet) of SSL itself still holds for realistic dataset distributions (that are usually not balanced at various degrees). This problem is important when applying SSL to real-world settings (with unknown distributions).  Differently, [3] is motivated to apply the common practice (SSL for model pre-training) to long-tailed recognition tasks.
> > > - ***How did we involve SSL?*** We conduct extensive studies to directly compare how SSL and supervised learning (SL) differ in the learned feature representations. We make the following observations that we believe are new and not presented in [3] or elsewhere: SSL performance is not affected by dataset distributions so much as SL; and SSL consistently produces balanced feature spaces for both balanced and imbalanced data distribution (Sec 3.2). We also dig deep into the balanced feature space learned by SSL, and show that such a feature space gives better generalization ability both in long-tailed setting (Sec. 3.3) and balanced setting (Sec. 4.3).  In comparison, for long-tailed recognition, [3] only focuses on the classification performance of the model trained with SSL + SL.
> > > - ***How did we improve SSL?*** Based on our studies, we further propose KCL to combine the strengths of SSL and supervised learning, which not only substantially improves the model performance on long-tailed recognition (Sec. 4.2), but also improves generalization ability of the models learned on balanced models (Sec.4.3).
> > >
> > > Besides, we are glad to see that **the author’s clarification on their experimental setup indeed provides strong evidence for our claims made in our Response to AnonReviewer4.** We detail as follows.
> > >
> > > **1.**  As explained by the author,
> > > > “... the control variable is whether you use pre-trained weights or not, i.e., the only difference is starting from a pre-trained network rather than random.”
> > >
> > > The only difference between their model and the baseline is their model uses a SSL pretrained model while the baseline uses a randomly initialized network (i.e. training from scratch). The conclusion one can draw from that experiment comparison is **SSL pretrained models are better than randomly initialized models**, which is the most basic conclusion from any SSL literature [5]. That’s why it is necessary to make direct comparisons between SSL and supervised learning (eg. SSL 200 epochs + SL 90 epochs vs. SL 90 + 90 epochs as we did in our response), otherwise **there is no evidence to support “ssl pretraining gives additional benefits compared to supervised learning”.**  The statement that “SSL pretrained models are better than randomly initialized models” does not disprove our conclusion that “ssl pretraining gives no additional benefits compared to supervised learning”.
> > >
> > >
> > > Moreover, **the additional experimental results given by the author further proves our claim “the performance improvement observed by [3] may come from longer training”.** To see this, comparing SSP (200+90 SL epochs) with the baseline model (90 SL epochs), the improvement brought by SSP is 2.541 points (50.652-48.110). However, when the supervised learning extends to 200 epochs, comparing SSP (200+200 epochs) with the baseline model (200 epochs), improvement by SSP drops to 0.726 points (51.9 - 51.174). This trend is consistent with the known observation on the role of ImageNet pretraning [6]: when the training schedule is long enough, pre-training is not always necessary.
> > >
> > > Differently, in our experiments, we first tune the parameters of KCL to achieve similar performance as the SL trained on ImageNet. We then compare KCL with various baselines on long-tailed recognition (Sec 4.2) and normal downstream tasks (e.g, object detection) (Sec.4.3). Our  KCL achieves superior long-tailed recognition performance,  and gives better downstream task performance, demonstrating the superiority of our KCL method in obtaining more generalizable models.
> > >
> > > **Therefore, we argue that these nuances in training setups make the fair comparison and controlled experiment very important. We strongly believe our evaluation protocol is an appropriate one. We welcome further discussions.**
> > >
> > > [5] Goyal Priya, et al. "Scaling and benchmarking self-supervised visual representation learning." ICCV 2019.
> > > [6] He, Kaiming, Ross Girshick, and Piotr Dollár. "Rethinking imagenet pre-training." ICCV 2019.

---

> > > ### Author Response · Authors · 2020-11-18
> > > **Your clarification further confirms our claims: Part 2**
> > >
> > > **2.**
> > > > First, the technique in this paper … also requires label information in the first training stage. Therefore, this is essentially not an exact self-supervised learning technique.
> > > >  The consistent improvements already confirm the successful practice of combining supervised learning and SSL.
> > >
> > > This is not correct. First, it’s true that we need to use label information. However, **we NEVER claim that our proposed KCL is a self-supervised technique.** We only say that KCL is proposed to combine the strengths of self-supervised contrastive learning and supervised learning.
> > >
> > > Second, by combining the strengths of SSL and supervised learning, we mean **a representation model optimized by KCL can achieve being discriminative and balanced at the same time.** Our evaluation of KCL for both long-tailed recognition (Sec. 4.2) and for normal recognition tasks (Sec 4.3) clearly shows that KCL achieves this target successfully. However, your paper only shows that ssl pretraining is better than random initialization, which has been widely studied in various settings and accepted as a common belief in the self-supervised learning community.
> > >
> > > **3.**
> > > > “Second, to emphasize again, our paper has extensively tested … The consistent improvements already confirm the successful practice of combining supervised learning and SSL. …”
> > >
> > > First, we respectfully disagree that the experimental setup used in your paper is the most appropriate one. To prove that SSL is beneficial, it is certainly necessary to ablate the factor of longer training. In addition, **your paper only shows that ssl pretraining is better than random initialization, and does not investigate the properties of SSL ITSELF on representation learning.**
> > >
> > > Instead, our study in Sec 3, directly investigates self-supervised learning itself on multiple datasets with varying dataset distributions. To get a fair comparison, we make the following experimental designs: 1) Instead of using SSL as a pretraining technique, we directly use SSL to learn feature spaces to compare with the ones learned with supervised learning. 2) To make sure the only varying factor is representation models are trained with datasets of different distributions, we train the evaluation classifier on a common balanced datasets. Both the final results and our balancedness metric show that SSL performance is robust to dataset distributions and can consistently produce a balanced feature space. Such an empirical conclusion is core to our paper and helpful for understanding SSL better, which are not presented in [3]. We also did not see any experiments in your paper supporting this claim.
> > >
> > > **4.**
> > > > Finally, as Reviewer4 also noted, there is no performance advantage over our SSP method. ... should not be considered as the main contributions of this paper.
> > >
> > > In our view, **the experimental setup of SSP might be problematic**, and should not be used as a fair baseline in the literature of long-tailed recognition. In contrast, our KCL is evaluated with strictly the same setting as the existing long-tailed recognition methods, e.g., the number of training epochs of KCL is carefully tuned to align with baseline methods, making our KCL a valid and reliable SOTA.
> > >
> > >
> > > **5.**
> > > > ... we would like to emphasize that the contributions of past works should also be fairly treated and acknowledged ...
> > >
> > > We strongly disagree with this statement and our main contributions have not been published in [3] according to the discussions above.
> > >
> > > **6.**
> > >
> > > > ... it seems to be inconsistent with what was reported in the paper (e.g., -norm 46.7 in Table 2, while 48.1 here) ...
> > >
> > > The first one is directly quoted from the original paper, while the latter one is our re-implementation.

---

### Official Review · AnonReviewer2 · 2020-10-28
**Interesting Set of Experiments but Insufficient Clarity and Evaluations**

**Rating:** 5
**Confidence:** 5

**Review:**

**Overview:** The paper presents experiments showing that the contrastive learning losses produce better embeddings or feature spaces than those produced by using binary cross-entropy losses. The experiments show that embeddings learned using contrastive learning losses seem to favor long-tailed learning tasks, out-of-distribution tasks, and object detection. The paper also presents an extension of the contrastive loss to improve the embeddings. The experiments in the paper  use common and recent long-tail datasets as well as datasets for object detection and out-of-distribution tasks.

**Pros**:
*Interesting problem and approach*. I think the paper tackles a hard and important problem, i.e., learning from a long-tailed dataset. Overall, I think that learning a feature space improving the learning from these imbalanced datasets is an interesting idea.

*Clarity of the paper*. Overall the clarity of the paper is good. The motivation is clear and the narrative is clear overall. However, I think the clarity in the experiments is insufficient, see below.

**Cons**:
*Insufficient clarity in the experiments*. I have several concerns with the experiments:
1. The *balancedness* metric in Eq.(3) may not be a robust metric for measure performance. The reason I am not convinced about this metric is that if the accuracies of the classifier are low but equal, then the metric will say that the *balancedness* is good. I think a good metric for a classifier learning from an imbalanced dataset is one that indicates if the overall accuracy is high, maintains the many-shot the classification accuracy high, and increases the accuracy of the classes in the tail. I think this metric does not indicate if the overall accuracy is high.

2. I am not convinced about how classifiers are trained in experiments in Sec. 3.2. The paper trains and tests using a balanced set after learning a feature space. To my understanding, the challenge of learning from a long-tailed dataset is to test whether a classifier can generalize well for classes with few training samples while maintaining a good performance on classes with more training examples. Thus, by training a linear classifier with a balanced set using the learned feature space does not really comply with learning from an imbalanced dataset. I think if the experiments would've been stronger if they included results of a trained linear classifier on the learned embedding and still showing good results, then I would be more convinced about the impact of a contrastive loss. From the practical point of view, what matters is the classifier performance. In practice, it is challenging to have a balanced dataset as the paper used. The main question is about the performance when training a linear classifier from a long-tailed dataset using the learned representation.

3. Datasets derived from ImageNet-LT. While I value the goal of using different datasets varying the imbalance in a dataset, I am not convinced that ImageNet-LT is the dataset to use. The reason is that ImageNet-LT is a synthetic long-tailed dataset. In fact, while the dataset shows imbalance, it does not necessarily follow a power-law distribution. I think the generation of these datasets in the experiments should be done using a power-law distribution. From the text, it is unclear how the datasets from ImageNet-LT were generated for the experiments.


Minor concerns:
*Plots lack information*. What is the y-scale in Fig. 2? The figure is missing y-scale information and it is hard to interpret the gap in accuracy  between CE and CL in the left plot. Same comment for Fig 4, what is the scale in y-axis?

---

> ### Author Response · Authors · 2020-11-12
> **Response to AnonReviewer2: clarification for experiments in our submission**
>
> We clarify our experiment settings and results here and welcome further discussions.
>
> The reviewer might misunderstand one of our core contributions, the “balancedness” of a feature space. **Balancedness defined in Eqn. (3) is not used for evaluating the performance of a classifier but for measuring the quality of a feature space, i.e.,  whether it is skewed to some dominant classes and leads to biased classification accuracy.** This property is found by our extensive empirical studies on feature representation learning in various practical settings. When two feature spaces share similar discriminativeness, the more balanced one gives better generalization ability. This conclusion not only holds for long-tailed recognition (Sec. 4.2), but is also applicable to the normal balanced datasets (Sec. 4.3). So instead of only focusing on learning  “discriminative” feature space as in previous works, we propose “balancedness” as another important property for a good feature space and we suggest learning both discriminative and balanced feature spaces.
>
> **A1.** *[The balancedness metric may not be a robust metric for measuring performance.]*
>
> The reviewer might misunderstand the purpose of developing the balancedness metric as explained above. We would like to emphasize that the balancedness metric is only used to measure feature spaces from a new perspective, which is a complement to usual performance measures (e.g., accuracy), but not a replacement. We will explicitly clarify it in the revision.
>
> **A2.** *[I am not convinced about how classifiers are trained in experiments in Sec. 3.2.]*
>
> We agree with the reviewer on the evaluation of long-tailed classifiers and we have presented the experiments in Sec. 4.2. But **the reviewer misunderstands the study in Sec 3.2, whose purpose is to investigate how different loss functions (e.g., CE and CL) affects the balancedness of  learned embeddings, not to study the long-tailed recognition performance**, as explained multiple times in the introduction and Sec. 3.2,
>
> **<Experimental setting is Sec. 3.2 is reasonable>**
> To evaluate feature spaces from different losses, we first train several representation models with the six imbalanced datasets. However, the properties (balancedness and discriminativeness) of a feature space relies on the classification accuracies of learned classifiers. “Since linear classifiers are easily biased by skewed training dataset distribution, it is necessary to eliminate the imblancedness of the evaluation datasets for reliable representation evaluation. Thus, we use the (balanced) training and test sets of ImageNet to learn classifiers and evaluate their classification accuracy.” We have explained this in the first paragraph of Sec. 3.2.
>
> **<Long-tailed classifiers are evaluated by exactly following the setting you explained here>**
> For detailed settings and results for long-tailed recognition, please refer to Sec.4.2. It clearly shows that our proposed KCL loss substantially outperforms existing SOTAs, e.g., bringing 4.2 absolute points improvement on ImageNet-LT.
>
> **A3.** *[Datasets do not follow a power-law distribution.]*
>
> The ImageNet-LT dataset is generated following a power-law distribution as explained in [2]. Therefore, based on the equation generating our datasets (Eqn. (5)), it is clear that our datasets also follow power-law distributions. We also mention the datasets are generated by power law in the supplementary material.
>
> [2] Liu, Ziwei, et al. "Large-scale long-tailed recognition in an open world." Proceedings of the IEEE Conference on Computer Vision and Pattern Recognition. 2019.
>
> **A4.** *[missing y-axis in Fig.2 and Fig.4]*
>
> The y-axis is removed for clarity as we care more about the relative value. The precise values are all provided in tables. We will add a y-axis in the revision.

---

> > ### Comment · AnonReviewer2 · 2020-11-22
> > **I Understand the Balancedness in the Paper But Lacks Statistical Rigor**
> >
> > A1. **Balancedness defined in Eqn. (3) is not used for evaluating the performance of a classifier but for measuring the quality of a feature space, i.e., whether it is skewed to some dominant classes and leads to biased classification accuracy.** The definition is using classification performance (i.e., $a_i$) to evaluate the feature space and also depends directly on how the classifier was trained on. I think a more robust metric is to measure balancedness using class distribution properties and not via trained classifiers. For example, why not using a GMM to measure how the Gaussians PDFs  overlap per class? Or, how about using Fisher discriminant analysis? Just stating that the metric "was found by our extensive empirical studies on feature representation" brings more questions than answers and buttresses my point that the metric lacks statistical rigor and it is a heuristic. What experiments? How were they conducted? On what datasets? It would've been nice if the authors to included these "extensive" experiments.
> >
> > A2. Again, given my answer above. How can we trust Eqn. (3) if it lacks statistical rigor and depends directly on how the classifier was trained? I would be more convinced if the authors presented a statistical analysis demonstrating that the "Balancedness" metric produces a feature space where a Bayesian Classifier (which is the optimal classifier) performs well? How can this "Balancedness metric" can show or relate with the goal of minimizing the probability of misclassification?
> >
> > A3. While generation of ImageNet-LT uses a power law, the evaluation protocol does not make sense. My comment was more related to the evaluation of OLTR that divides many-, medium-, and few- shot classes with arbitrary thresholds. I think the paper is missing a plot like Fig. 6 in OLTR paper.
> >
> > A4. Removing information from a plot will never make it clearer, especially the units and scale y-axis is crucial to judge appropriately the performance of the method.

---

> > > ### Author Response · Authors · 2020-11-23
> > > **More explanation on balanced feature space and long-tailed recognition evaluation [1/2]**
> > >
> > > We appreciate the valuable feedback given by AR2. However, we still noticed some misunderstandings on the definition of balancedness and the common evaluation protocol for long-tailed recognition, due to our unclear presentation. Thus, we clarify them here.
> > >
> > > Balancedness is not for
> > > > “minimizing the probability of misclassification”
> > >
> > > Or it is not for evaluating whether the features from different classes form clearly separated clusters. Thus, the GMM and LDA-alike methods, which are more for inspecting whether features from different classes form clear clusters, are not directly applicable here. Below we give detailed explanations.
> > >
> > > **On what is a balanced feature space.** Balancedness is used for measuring “uniformity” of the feature distribution for a feature space. If the features from different classes are uniformly distributed within the space, even though they are mixed together and not well separated, this feature space is still balanced. The classification accuracies on different classes within this feature space are similar to each other. Correspondingly, the imbalanced feature space is the one that some classes form dominatingly larger clusters than other classes, with much higher classification accuracy than others.  See Fig.4 (right) for an intuitive illustration. MoCo (the blue line) gives relatively similar classification accuracies on all classes, yielding a horizontal-like line. The underlying feature space is balanced.  In contrast, the class-wise accuracy of supervised learning (the green line) for some classes (with more training instances) is higher than the other classes, producing a skewed line in the figure. The underlying feature space is not balanced. One feature space is seen to be totally balanced if its classification accuracy curve is a horizontal line in such a figure. Fig.4 (right) clearly shows the superiority of contrastive loss (SSL) at producing a balanced feature space. It is noteworthy that balancedness is complementary to “discrimiantiveness”, which is directly related to classification accuracy. Balancedness is more for model’s generalization performance as we explained in Sec 3 and 4.
> > >
> > > **On why use the current balancedness metric.** From the above observations, we define balancedness as the “variance” or “difference” of the accuracy across different classes. Our developed balancedness metric is informative enough for inspecting the above feature space property. Though it is not the only way and there may be better alternatives, the current metric is sufficient for serving the purpose and supporting our main contributions.
> > >
> > > **On why not use LDA/GMM.** Regarding LDA/GMM, as mentioned above, they are NOT designed for modeling how uniform the distribution of different-class features within the space is. Moreover,  we indeed tried LDA before, aiming to use the within-class and between-class scatters to inspect the balancedness of feature space. We apply supervised learning to train the models and obtain the features. Then compute their projected within-class ($d_i$) and between-class ($d_{ij}$) distances. The results are listed below (on ImageNet-LT).
> > >
> > > |split|$d_i$|$d_{ij}$|$r=d_{ij}/(d_i+d_j)$|
> > > |--|--|--|--|
> > > |All|14.74|9.43|0.32|
> > > |Many|14.75|9.79|0.33|
> > > |Medium|14.94|9.17|0.31|
> > > |Few|14.09|9.37|0.32|
> > >
> > > Unfortunately, we found such LDA criterion is not very informative even for the feature space learned from supervised learning. There is no noticeable difference between head classes and tail classes in terms of $r$ when directly investigating their features,  even though there is a huge gap between their classification accuracies. This is probably because such metrics are not suitable for high-dimensional feature spaces (2048 dimension in our case).
> > >
> > > Therefore, we abandoned LDA and turned to investigate the classification accuracies and tried the following ways.
> > > - STD: The standard deviation of accuracies
> > > - ENT: The entropy of normalized accuracies
> > > - UNI: The uniformity of accuracies (our submission)
> > > Among them, we found UNI performs well and is able to provide a relatively appropriate description of balancedness.  Both STD and ENT are easily affected by perturbations, and can not reflect the positive correlation between accuracy and the number of training instances of the class.
> > >
> > > In the submission, we do not provide a detailed explanation of the above investigation, because **compared with how to define balancedness, our other contributions are much more important.** Besides the developed balancedness metric, our accuracy visualization (Fig. 4) and generalization experimental results on both imbalance setting (Sec 3.3) and balanced setting (Sec 4.3) also provide strong evidence for understanding the superiority of balanced feature space.  **Moreover, we would like to emphasize that our main contributions as highlighted at the end of the introduction and our [response to AR3](https://openreview.net/forum?id=OqtLIabPTit&noteId=kXuA-zDzf6u) should not be neglected!**

---

> > > ### Author Response · Authors · 2020-11-23
> > > **More explanation on balanced feature space and long-tailed recognition evaluation [2/2]**
> > >
> > > > I would be more convinced if ... the "Balancedness" metric produces a feature space where a Bayesian Classifier (which is the optimal classifier) performs well?
> > >
> > > We would like to emphasize that the linear classifier is commonly used in modern deep learning based classification problems (known as linear protocol)[1], and widely adopted and acknowledged by the self-supervised learning community. By strictly controlling the variable (different datasets to learn representation) and choosing a common balanced dataset to learn the classifier, it is able to reveal the discriminativenss and balancedness of different feature spaces. Bayesian classifiers are not so widely used as  linear classifiers in modern DL literatures for feature quality evaluation. Even if we use a Bayesian classifier for evaluation, how to train it (estimating its parameters/probability densities)  and on what dataset are still not clear.
> > >
> > >
> > > > I think the paper is missing a plot like Fig. 6 in OLTR paper.
> > >
> > > The Fig.6 in OLTR is used to visualize the F1 score, which is a metric for open-set recognition and is not relevant to long-tailed recognition performance evaluation.  As for the visualization format, our Figure 4 gives a similar one showing the accuracy for every class and the classes are sorted from head (left most) to tail (right most).  There are 1000 classes in total. For clearer visualization, we divide them into 10 bins according to their training instance numbers. Each dot in the figure represents the average accuracy of the corresponding bin. Fig.4 is able to provide a straightforward and intuitive demonstration of how a balanced feature space works. It clearly shows our proposed method outperforms all the other baselines and brings consistent accuracy improvement over all the classes (including both head and tail classes). We are happy to provide such visualization for all models trained in the study section (Sec.3) in the revision.
> > >
> > > >  the units and scale y-axis
> > >
> > > We have updated the figures, please refer to the revised submission.
> > >
> > >
> > > If you have further questions, please let us know.
> > >
> > > [1] He, Kaiming, et al. "Momentum contrast for unsupervised visual representation learning." Proceedings of the IEEE/CVF Conference on Computer Vision and Pattern Recognition. 2020.

---

### Official Review · AnonReviewer1 · 2020-10-28
**Interesting paper with some improvements possible**

**Rating:** 6
**Confidence:** 5

**Review:**

In this paper, the authors propose a new loss function to learn feature representations for image datasets that are class-imbalanced. The loss function is a simple yet effective tweak on an existing supervised contrastive loss work. A number of empirical tests are performed on long-tailed datasets showing the benefits of the proposed loss in beating state of the art methods.  Some specific questions are listed below:

1. Is Figure 1 hypothetical ("desired outcome") or real (based on actual observations)?

2. Minor style comment: please don't call your own contributions "important" :-) - that is for others to decide.

3. Eqn. (3) is not clear at all - please provide an intuitive explanation and motivation. It seems to appear out of thin air.

4. Is there a tradeoff between accuracy and balancedness? Figure 2 seems to suggest so. For example, if we did not train CE loss to maximal accuracy, would be automatically get the balancedness property? Was this tested? How about a combined loss CE + CL to get the best of both worlds? And similarly, how about CE + KCL?

5. Table 1 the delta for VOC seems to be computed wrongly.

6. Figure 7: it's strange that there are no numbers on the y-axis.

7. What is the ImageNet accuracy of KCL?

---

> ### Author Response · Authors · 2020-11-12
> **Response to AnonReviewer1: replies to the questions and suggestions**
>
> **A1.** Figure 1 is a hypothetical illustration of the feature spaces learned with different loss functions. However, this illustration is strongly evidenced by the experimental results on real large-scale datasets. As shown in Fig.4 (right):
>  - SL (using the CE loss) gives promising overall performance but heavily skewed accuracy curve, indicating the learned feature space is discriminative for different classes but biased to the head-classes;
>  - CL (MoCo) gives the poorest overall performance but a balanced accuracy curve, i.e., the learned feature space is well balanced but not so discriminative.
>  - Our proposed KCL performs the best, presenting the strengths of both CE and CL. This implies its learned feature space is both balanced and discriminative.
>
> **A2.** Noted with thanks. We will revise the paper accordingly.
>
> **A3.** Eqn. (3) defines a measure of the balancedness of a feature space based on the following intuition. As explained in the context of Eqn. (3), a feature space is balanced if the linear classifier trained on its sample features does not give biased performance to any classes, i.e.,  the classifier gives balanced (similar) accuracy on all the classes (“similar degrees of linear separability”). Since balancedness (similarity) of a set of values can be computed by their pairwise Gaussian distance [1], we use Eqn. (3) to compute the balancedness of the class-wise accuracy of a linear classifier and take it to measure the feature space balancedness. We will make this clear and add a more intuitive explanation in the revision.
>
> [1] Wang, Tongzhou, and Phillip Isola. "Understanding Contrastive Representation Learning through Alignment and Uniformity on the Hypersphere." arXiv preprint arXiv:2005.10242 (2020).
>
> **A4.** *[Is there a tradeoff between accuracy and balancedness?]*
>
> We would like to clarify that we are proposing balancedness as a complementary property to discriminativeness (or classification accuracy) for representation learning. There is no trade-off between them. More detailed explanations are given below.
>
> *[More explanation for Fig.2.]*
> For Fig.2, by comparing the performance (left) and balancedness (right) curves given by CE loss (green curves), we can observe that the performance increases as balancedness improves, which demonstrates a positive correlation, instead of trade-off,  between accuracy and balancedness. When comparing CE loss with CL loss,  CE gives high accuracy but low balancedness, while CL gives poor accuracy but strong balancedness. But this does not imply there is a trade-off between accuracy and balancedness because these two loss functions focus on different aspects of feature learning (one for discriminativeness, the other one for balancedness). Therefore, it is possible to optimize discriminativeness and balancedness simultaneously, which motivates us to propose the KCL loss.
>
> *[Would a random model automatically get the balancedness property?]*
> Yes, it is possible that a random classification model gives optimal balancedness property. Based on Eqn. (3), all zero accuracies give the largest balancedness score.  However, please note that balancedness is a complementary property to discriminativeness.
>
> *[How about a combined loss CE + CL?]*
> We have tried to train a model with a combined loss CE + $\lambda$ CL with $\lambda$ as the weighting factor.  **Our experiments reveal that directly optimizing these two losses gives sub-optimal results.**  When the CL term weighting factor increases (learned more balanced representations), accuracy decreases. The experimental results on ImageNet-LT are given as follows
>
> | $\lambda$ | Many  |  Medium |   Few   |   All |
> | --------------- | --------  | ------------ | --------  | ---- |
> | 0.0 | 59.735 | 45.113 | 27.082 | 48.110 |
> | 0.3 | 59.326 | 44.142 | 28.408 | 47.744 |
> | 1.0 | 58.169 | 42.151 | 24.462 | 45.744 |
>
> However, this does not mean that balancedness and good performance cannot be achieved at the same time. Actually, KCL is motivated by achieving these two properties simultaneously. **The above results show this target is non-trivial, further validating the novelty and necessity of our KCL loss.**
>
> *[How about a combined loss CE + KCL?]*
> The results can be inferred from the CE and KCL results. (1) KCL is designed to combine the strengths of CE and CL, which gives both good balancedness and accuracy. (2) KCL + CE loss will achieve the performance lying between  CE and  KCL results.
>
> **A5.** Thank you for pointing it out. We will address it in the revision.
>
> **A6.** There is no Fig.7 in our paper. The reviewer may refer to Fig.2 and Fig.4. We remove the y-axis value for more clearly showing the relative difference of different methods. The actual values of Fig.4 are all provided in Table 2. We will add y-axis values in the revision.
>
> **A7.** KCL achieves an accuracy of  76.814 on ImageNet, as given in Table 5, which is comparable with the result of CE (76.616).

---

### Decision · Program_Chairs · 2021-01-07
**Final Decision**

**Decision:**

Accept (Poster)

**Comment:**


 This paper studies the difference between cross-entropy and contrastive learning losses in the feature representations that they learn, specifically looking at class-imbalanced datasets. The authors show that contrastive losses result in a more "balanced" representation, as measured by the balance of accuracy across the classes when a linear classifier is learned mapping from the feature representation to the class labels. They also show that empirically this tends to result in better generalization to downstream tasks. Inspired by this, they devise a simple modification of the prior supervised contrastive loss method and show that it can improve performance on ImageNet-LT and even generalization performance when trained on balanced datasets and applied to downstream tasks.

  The reviewers identified several weaknesses, including some clarity issues (R1), limitations of how balancedness is measured and lack of theoretical/statistical rigor in terms of the resulting claims (R2), and differences with respect to concurrent work (R4). A lengthy discussion occurred between reviewers and authors, as well as input from a co-author of the concurrent work. In the end, the reviewers were not fully satisfied both in terms of the balancedness measure and relationship to the concurrent work.

  Overall, despite this and the valid limitations of the work, I recommend accepting this paper as I believe the contributions outweigh the limitations, and that the findings would be interesting to the community. First, the paper provides some interesting analysis of balancedness and differences across these two loss functions, as well as connections to generalization, which even the concurrent work does not provide. The resulting method, while being a simple modification of the supervised contrastive loss work, is effective both for long-tailed datasets and generalization to downstream tasks (even when trained in a balanced manner) which is nice. In the end, we should not use [3] to reject this paper since it was accepted right before the ICLR deadline.

  However, I **strongly** recommend that the paper address the valid limitations mentioned in the discussions. Specifically:
  1) While I agree that [3] is concurrent work, this paper should none-the-less tone down its claims of being the first in exploring balance for the camera-ready version and clearly address differences between this paper and that one (even if mentioned as concurrent work). It is important to give credit when it is due, and while I think [3] is a different perspective it should be mentioned. Further, the claim that their methodology is not correct is highly arguable, so this should not be mentioned; rather the differences in perspectives and what each paper shows should be emphasized. Even without [3], self-supervised pre-training (initialization) should arguably be included as a baseline given that it is the logical first choice for incorporating self-supervised learning.

  2) Like R2, I do not believe the balancedness metric shows uniformity of the feature space. This would have to be shown through methods such as t-SNE or in some other way. Being linearly separable in a balanced way across classes (which is what you showed) is not sufficient to show that feature space "uniformity". One can draw many feature space distributions that do not have the intuitive meaning of this (which isn't precisely defined by the authors) but still be linearly separable. I recommend authors remove this type of characterization (unless they can define/show it) and instead include a discussion of the limitations of the current methodology for measuring balancedness. Figure 1 should also emphasize that it is notional (not from real data).